# A Focus Group Interview Study of the Experience of Stress amongst School-Aged Children in Sweden

**DOI:** 10.3390/ijerph17114021

**Published:** 2020-06-05

**Authors:** Alexandra Warghoff, Sara Persson, Pernilla Garmy, Eva-Lena Einberg

**Affiliations:** 1Faculty of Health Sciences, Kristianstad University, 291 88 Kristianstad, Sweden; alexandra.warghoff@gmail.com (A.W.); Sara.persson90@hotmail.com (S.P.); evalena.einberg@hkr.se (E.-L.E.); 2Clinical Health Promotion Centre, Medical Faculty, Lund University, 221 00 Lund, Sweden

**Keywords:** stress, school-aged children, experience, focus group, interviews, qualitative study

## Abstract

The study explored experiences of stress in children aged 10–12. An inductive qualitative design was used. Ten focus group interviews were conducted with Swedish schoolchildren (*n* = 42) aged 10–12. The interviews were audio-recorded, transcribed verbatim, and analyzed using qualitative content analysis. The study identified one theme—“Body and mind react”—and three subthemes: (1) Friendships matters, (2) Parental stress affect, and (3) Trying to do my best. Stress often occurred in the children’s everyday environment. The children had experienced how stress could be expressed both physically and mentally, inside and outside school. The children also had the experience of seeing people in their environment being stressed, which could create feelings that affected the children. The experience of the children was also that stress can sometimes be good. Stress related to peers, family, school, and society was commonly experienced by children aged 10–12. Both positive and negative stress was reported.

## 1. Introduction

The life course relevance for stress in children follows from the results of longitudinal studies showing that stressors and adverse environments during childhood are associated with elevated risks for developing depression and anxiety disorders during adulthood [1,2]. Stress in children is linked to the societal context as well as individual factors. Socioeconomic factors such as the parents living together, parental employment, housing, and education affect stress levels in children [3]. Psychosomatic complaints in children are often associated with stress [4], defined as a nonspecific mental or somatic result of any demand upon the body [5]. Stress can be manifested as physical symptoms such as stomach pain, headache, dizziness, nausea, or a throbbing heart and as mental symptoms such as nervousness, sadness, depression, or anger [6]. The research question for this study was: How do school-aged children experience stress?

### Background

The ages 10–12 are defined as early adolescence [7]. They are characterized by physical, emotional, social, and cognitive milestones, such as increased coordination and strength. Puberty may begin during these years. The ability to interact with peers and to engage in competition increases. Beliefs and values that will guide present and future behaviors are being developed and tested. Children in this age have a strong group identity and increasingly define themselves through peers [7]. Having conflicts or being bullied at this age can cause massive stress [8]. Children at this age are acquiring a sense of accomplishment based upon the achievement of greater strength and self-control. However, body changes can also cause stressful emotions regarding body image [9]. Self-concept is related in part by success in school. Early adolescents have an increased ability to learn and apply skills. In these years, abstract thinking begins; however, this thinking reverts to concrete thinking under stress. Though abstract thinking starts in this age period, children at this age are still developing the method of reasoning and are not able to make all the intellectual leaps, such as conducting hypothetical reasoning [7]. This is also the time when children learn to expand their way of thinking beyond their personal experiences and knowledge and start to view the world outside of an absolute black-white or right-wrong perspective. The ability to interpret and understand cause-and-effect sequences develops during this period of early adolescence. Early adolescents can answer who, where, and when questions, but still may have problems with why questions [7]. In prepubertal children, depression is less common, but increases with age. Girls experience increased stress and anxiety, which are often linked to schoolwork; performance anxiety leads to poorer well-being [10].

Stress is an everyday challenge for school-aged children and is something that cannot be avoided. However, stressors can either facilitate positive development or result in traumatization depending on the art, intensity, dose, and time period, as well as the child’s coping strategies [11]. The individual’s perception of the stressor is influenced more by coping strategies than the stress event itself [12]. Furthermore, according to Bronfenbrenner [13], a child’s development is influenced by different systems that surround that child at various levels, interacting either directly or indirectly. The family and the school are micro-systems and, through the relationships between them, form mesosystems. Environments that indirectly affect the child are the parents’ work situations, relatives, and friends, which form the exosystem. At the macro level, norms and values in society affect the child’s situation [13]. The different systems and ways of interacting with each other affect children’s life situations, including stress-related experiences.

Results from the Swedish contribution to health behavior in school-aged children showed that 18% of the children aged 11 found school to be stressful, compared with 36% and 61% of the 13- and 15-year-olds [14]. Brobeck, Marklund [15] found that children can also experience stress as positive if it pushes them forward and that stress can motivate extra effort with, for example, schoolwork.

## 2. Purpose

Given the challenges faced by the age group of early adolescence (10–12 years), and that values and beliefs are formed in early adolescence, there is a need for studies listening to the voices of children. We have found few earlier studies interviewing children in this age group about their experience of stress [15,16]. Thus, there is a need for more recent research on the subject. This study aimed to explore experiences of stress in children aged 10–12.

## 3. Methods

### 3.1. Design

An inductive qualitative design was used to elucidate how the children experienced stress [17]. The inductive approach is data-driven and works from the specific to the general, searching for patterns. It is preferred when knowledge in the field is sparse [17]. Focus group interviews were conducted to achieve a broad understanding of the phenomenon (stress) by interacting with a group of children [18]. The focus group methodology has been proven useful and is appreciated in research with children [19].

### 3.2. Sample

A strategic sample of two schools in two municipalities in southern Sweden was used. One school had an urban catchment area with children with higher unemployment rates among the parents and more immigrants as compared to the average for Sweden, while the other school had a catchment area of a rural area with a population with average employment rates among the parents and a population of mainly Swedish origin [20]. The final sample consisted of 10 focus groups with a total of 42 children aged 10–12 (22 girls and 20 boys), see Table 1. The intention was to conduct focus group interviews with four to six children in each group. This was the case in eight focus group interviews. However, in two groups, there were only two and three children.

### 3.3. Measures

Qualitative data were collected using semistructured focus group interviews. Each focus group interview was conducted by one moderator and one observer with experience talking with children in this age group. All authors alternatively played the roles of moderator or observer. As registered nurses, two authors (A.W. and S.P.) were master’s students, and two (P.G. and E.-L.E.) were pediatric nurses with extensive experience as school nurses and with encounters with children. The semistructured interview guide [21] consisted of open-ended questions regarding stress. Examples of questions were, “What do you think of when you hear the word stress?”, “Can you please describe situations when stress was good”, “Can you please describe situations when stress was bad?”, and “How do you notice that someone is stressed out?” To obtain an informative and deep understanding, questions such as “Please tell me more” and “What do you mean?” were posed. Two pilot interviews were conducted at the urban school to evaluate the interview guide. No changes to the interview guide were necessary; the pilot interviews were of good quality, and they were included in the analysis. The focus group interviews were conducted during school hours at the school during May and September 2019. Observer notes (memos) were reviewed with the moderator after each focus group interview was completed. All focus group interviews were audio-recorded, lasted 30–45 min, and were transcribed verbatim.

### 3.4. Analytic Strategy

The transcribed texts were analyzed using qualitative content analysis [22]. This is considered an appropriate approach for analyzing interview data and systematically interpreting its meaning by focusing on relevant data. Content analysis focuses on variations in respondents’ experiences [23].

The analysis started with reading each focus group interview transcript to obtain a comprehensive sense of the overall situation. This was performed independently by A.W. and S.P., who then met and discussed the content of the text. Subsequently, all text was read again to identify meaning units related to the aim of the study. Together, two authors (A.W. and S.P.) condensed the meaning units, and then abstracted and labeled them using a code, considering the text as a whole. Regular discussions with another author (E.-L.E.) were held throughout the analysis of the meaning of those units and codes. The codes were compared based on their differences and similarities, and they were then sorted by all authors into three subthemes under a single theme. All authors regularly reviewed and reflected on the analysis process and the coding frame. Trustworthiness was increased because the analysis was discussed among all four authors until consensus was obtained. Quotes from the interviews are presented in the results section to illustrate the theme and subthemes.

### 3.5. Ethical Considerations

The study was approved by the Lund Regional Ethics Review Board (EPN 2018/842). The Helsinki Declaration’s ethical principles and the basic ethical principles of autonomy, justice, not to harm, and do good [24] guided the study. Letters with information about the study were sent to the principals of the two schools. After principal approval, written information about the study and its voluntary nature was sent to the children and their parents/legal guardians. Verbal assent was obtained from the children, and written informed consent was obtained from the parents/legal guardians.

## 4. Results

The study identified one theme—“Body and mind react”—and three subthemes: (1) Friendships matters, (2) Parental stress affect, and (3) Trying to do my best, as seen in Figure 1. The results of the present study revealed that stress was something that often occurred in the children’s everyday environment. Children’s experience with stress involved various factors that caused stress to arise and how stress could be expressed both physically and mentally. Stress was something that all the children had experienced both inside and outside of school. The children also had experience with seeing people in their environment being stressed, which could create feelings that affected the children. Additionally, the children’s experience was that stress can sometimes be good.

### 4.1. Theme: Body and Mind React

When the children talked about stress, they started to discuss how it could feel inside. The children explained that the mood was affected in several ways and that their emotions were negatively affected if they felt stressed. When stress took over and became more evident, the children felt that they were close to crying and could more easily break down at any time:

*"Then when you are stressed it sometimes feels like you can break down for kind of anything"* (Focus Group No. 10).

*“The one who is stressed does not want to do much, just wants to be for herself. Gets irritated easily. I think you can notice a little that he or she is sad. That person is not so vibrant, he or she is more by herself. He or she does not want to talk about what he or she might be sad about”* (Focus Group No. 2).

When the children heard the word stress, they got a feeling in their body that something was not good, that something was going to happen. A feeling of nervousness could appear, and the body could react by shivering. The mood could affect the children in that they became more tired, easily excitable, and angry about minor things.

*“If you are stressed, you are kind of very tired. You don’t want to talk to anyone. Just do what you must” * (Focus Group No. 5).

The children said that stress was usually painful and that they reacted by hurrying more than usual. This could result in them forgetting things that they would remember if they were not stressed.

#### 4.1.1. Subtheme: Friendship Matters

The children said that friendship mainly reduced stress, but when there were conflicts and misunderstandings with peers, stress increased. Confidence and security were things the children considered important in a friendship relationship. Safety in the form of friendship was something that the children perceived as important both inside and outside school. Likewise, having friends to come back to after every summer vacation could feel safe. The children said that trust in one another, in which secrets could be shared between classmates, was important. However, it was not always easy to keep secrets:

*“I get stressed if I’ve told a secret and then I realize that secret I probably shouldn’t have told”* (Focus Group No. 6).

Receiving unpleasant remarks or having their friendship harassed by one or more individuals were some of the situations experienced by children that can be considered stress factors, and potentially cause the children to withdraw. If the children were on bad terms with someone, they could not relax until the problem was resolved and the friendship was restored.

*“When you fight because both get angry at each other and you are angry at each other then I get stressed. When the break is over and we haven’t straightened it out then I get stressed because I might be sitting next to her in class”* (Focus Group No. 1).

The children had experiences of stress when they changed schools and were new to the class. Being bullied at school could create stress. This was something the children thought about even outside school hours and sometimes when they went to bed at night.

*“I get stressed when someone says, ‘You’re ugly’. Then I panic. I get angry and then you get depressed. I notice it when you don’t talk so loud and you become quiet”* (Focus Group No. 3).

The children also talked about the stress of having the sense that they had said or done something that they were not proud of and that they regretted. This was considered very stressful until the issue was resolved. The children said that sometimes it was not possible to solve the problem and that they experienced this as very stressful.

#### 4.1.2. Subtheme: Parental Stress Affect

The children had experienced the stress of other people in their environment, such as their parents. The children said that, when their parents were stressed, they noticed a change in behavior, such as forgetfulness, faster movement patterns, raised voices, anger, and irritation. The children also said that their parents could sometimes be angry, which created a feeling in the children that they themselves were the reason for their parents’ behavior.

*“When mom and dad are stressed and irritated, it feels like it’s my fault. I can get a bit annoyed and a bit sad because I think it is my own fault”* (Focus Group No. 2).

The children felt that their parents’ stress was usually about getting to work or a party on time, but there were also more pervasive stressors, such as stressful life events like divorce, grief, risk of unemployment, or worries about the economic situation. The children talked about factors outside of their control, such as their parents’ work situation or the death of a relative, which created stress in their parents, and which were difficult to deal with.

#### 4.1.3. Subtheme: Trying to Do My Best

The children discussed the pressure they felt when they were trying to do their best, both inside and outside school. The demands came from themselves, but they also felt pressure from school. Some of the demands were possible to deal with, while others were not. Many children said that the first thing they thought of when they heard the word stress was homework. The children expressed that their homework should be given priority, but that school performance could also create stress. According to the children, stress due to homework could cause problems in providing the information they needed to learn. If the children felt that they had too much homework, they could perceive that they had too much on their minds. The children expressed disappointment in themselves and said that they were saddened when they had a desire to perform well in all subjects. Studying before a test could create stress; the children could also experience residual stress after the test was completed and was being graded. The children said that they wanted good grades, not only for themselves but also for their parents and teachers.

*“You have to pass all subjects and get good grades and if your parents get angry that you have bad grades, you are very stressed because you want good grades”* (Focus Group No. 3).

The children had heard that grades were not important at their age, yet they felt pressure to succeed.

*“They say that it is not very important with grades yet, but it may feel a bit like it is very important”* (Focus Group No. 8).

Some children thought that there was a very different degree of difficulty in homework and tests at each new grade level and did not feel prepared for that change.

The children talked about getting to places and events on time. They said that they lacked time and that they had too much to do in too short a time. They had activities and they also wanted to hang out with friends.

*“You have a lot to do and a lot of things to do all the time and then you get stressed because you want to catch everything in time”* (Focus Group No. 7).

Although stress was often negatively experienced, the children also described positive aspects of stress. They said that stress could motivate them. It could even be fun to be a little stressed during a test, as this would compel them to get better grades. Stress could also create more focus and concentration, which the children felt would be beneficial. They believed that stress was what caused them to do their homework. Stress was also positive in that it encouraged the children to arrive on time for activities and school and that it would enable them to survive in an emergency.

*“Sometimes it’s good to be stressed, but sometimes it’s pretty bad because sometimes stress puts a lot of pressure. And sometimes, well … for example, if I must go to school, then I hurry and then it goes faster if I stress. So, it can be good”* (Focus Group No. 8).

## 5. Discussion

The children aged 10–12 described their experiences of stress, both positive and negative. Stress affected their minds and bodies, and they felt stress related to peers, parents, school, and leisure activities.

Friendship was a main topic of discussion, and having reliable friends was crucial for dealing with stress. However, when the friendship was in trouble, or if harassment and bullying occurred, the stress escalated. Bullying is linked to psychosomatic symptoms in children [25]. Children and adolescents who are victimized by bullying use analgesics to a higher extent than their peers [26]. The importance of having friends to help cope with stress and also conflicts with friends were added to children’s narratives about stress in a study conducted in 2007 [15]. In a recent study [16], the researchers found that social relationships are important for children’s stress-related experiences as these relationships can be described as “navigating the social minefield.” Therefore, school-based interventions aimed at promoting a friendly climate and preventing bullying are crucial [27,28]. School-based efforts to promote positive mental health and resilience to stressors have been found to be effective [29,30].

The children felt that the demands of school can create stress. In the present study, the children said that grades, homework, and tests were important for their future. Despite their young age, they focused on getting good grades, for their own sake and for that of others. The children experienced stress in connection with school assignments due to a perceived lack of time. Brobeck, Marklund [15] found that children can experience stress when they have too much to do and think about at the same time. The children in the study of Brobeck, Marklund [15] said that they had recreational interests that they thought were fun and that it was sometimes difficult to know what to prioritize when they had a lot of homework. A feeling of nervousness about not being able to do enough with school assignments could arise, even when the children knew that they had the capacity to do so. The children tried to keep the same pace in their math book as their classmates so that they did not seem worse than the others [15]. However, in the study by Brobeck and Marklund [15], the children did not indicate that their grades caused stress, but in our study, they did. The difference can be explained by an earlier introduction of grades in Sweden in recent years. Grades have been widely debated in society and the media, possibly raising awareness among children of how grades affect their futures. For children today to have a well-functioning and structured everyday life, they must be supported by school staff and guardians. For the children to feel that they have time for leisure activities and schoolwork, additional study time at school could be introduced. In this way, the children would not have to perform school tasks at home to the same extent. Also, all children may not receive the help they need at home, which may further increase their stress. Based on Antonovsky’s theory of the Sense of Coherence [31], manageability in this context can be about helping children reduce stress and achieve a functional everyday life. People in the children’s environment are resources that they need in and out of school. It is important for the children to feel that they have control over the demands they face, both now and in the future. For the children to feel that they can handle their situations, they must be able to cope with them both physically and mentally.

Stressed parents can create stress, according to the children. Parental stress could be about anything from arriving on time for an event to a divorce that completely changes the family’s circumstances. The children could experience negative conscience feelings when they were afraid that their parents’ behavior was due to something the children did. According to Vanaelst, De Vriendt [3], children’s feelings of stress can be affected by several events in their everyday environments, including both school and family. Children can also be affected by, for example, divorce when one of the parents experiences an increased burden and financial troubles. The children in the current study wished that they could help reduce their parents’ stress but said that it was difficult to do so when the parents had to perform the measures themselves. According to Brobeck, Marklund [15], children felt that their parents quickly became angry in times of stress; at such times, the children could feel that they were in the way. According to Vanaelst, De Vriendt [3], children’s development can be adversely affected by constant exposure to stress. Children’s physiological and psychological health is greatly affected by the stress they are exposed to from various angles; this can continue into adulthood and lead to depression [3].

Sheidow, Henry [32] showed that strong family functioning protects adolescents from the negative psychological impact of stress. The researchers concluded that prevention programs promoting family structure, routine, organization, and coping may reduce the risk of internalizing outcomes in vulnerable youth [32].

Results from the present study and other studies [15,16] about children’s experiences of stress can be understood with Uri Bronfenbrenner’s ecological systems theory [13]. Children’s narratives show that their stress-related experiences can be related to interactions with the surrounding people and environment, such as friends, parents, and school. Bronfenbrenner´s ecological system theory illustrates the ways in which children are influenced by different systems that surround them. At the micro-level, friends can cause stress via harassment or are reliable friends from whom the stressed child can get help to deal with stress. At the exo-level, the parent’s work situation can affect the relationships in the family and cause stress, and at the macro-level, political decisions, for example pressure to get good grades, can affect a child’s stress while in school. For adults encountering children who express stress, a system perspective can provide a broader understanding of children’s situation.

## 6. Strengths and Limitations

The strengths of the study are the rich data provided by 42 children in 10 focus group interviews. The benefit of focus groups is that they provide the possibility of getting further into a subject due to the interaction. The prerequisites are that the atmosphere must feel safe for the participants. For children aged 10–12, peers are very important. The discussions went well in all 10 focus groups; however, we do not know if we would have gotten other results through individual interviews. The purpose was to have four to six children in each focus group; however, in two cases, only two or three children were present, and they wanted to take part in the focus group interviews. These discussions went well and were rich; therefore, they were included in the total sample. The interviews were conducted at two schools in southern Sweden, and the questions about transferability remain.

## 7. Implications for Practice

The health and well-being of a child affects learning in school. Conversely, learning affects health [33]. Good academic achievement improves self-esteem and is related to mental health [34]. Positive experiences at school during childhood are associated with less job stress during adulthood [35]. Schools have important roles in providing children with knowledge and the abilities to establish a healthy life in the future. Therefore, school health professionals must conduct both preventive and promotional health activities while supporting the educational goals of the children. At the micro-level, stress can be reduced if the school health team jointly submits proposals for planning tests, homework, and schoolwork. For example, tests can be evenly distributed rather than several falling on the same week. Health conversations can include parents in a general dialogue about stresses on children and how families can reduce them [36]. At the meso-level, bullying can be prevented and a friendly school climate can be promoted by establishing and implementing antibullying policies through a partnership between school administrators, teachers, children, parents, and school health professionals [37,38]. At the exo-level, parental stress should be given attention as well because it affects children. If support to the parents can lead to reduced stress, it could also have a positive effect on the children. Parental academic involvement is important for academic achievement, but family stresses can limit this involvement at the adolescent stage [39]. At the macro-level, before making decisions affecting the everyday lives of children, the children should be asked questions that concern them [40,41]. A child’s view can contribute to a greater understanding of how issues affecting them can be resolved. Therefore, the child’s perspective should also guide efforts at the micro-level and is the reason why the school health team, in their efforts to help children with school-related stresses, should discuss possible solutions with those children before actions are taken.

## 8. Conclusions

Stress often occurs in children’s everyday environments. Stress related to peers, family, school, and society are commonly experienced by children. The children experience how stress can be expressed both physically and mentally, both inside and outside school. The children also had the experience of seeing stressed people in their environment, which could create feelings that affected the children. Children also reported that sometimes they experienced good stress.

## Figures and Tables

**Figure 1 ijerph-17-04021-f001:**
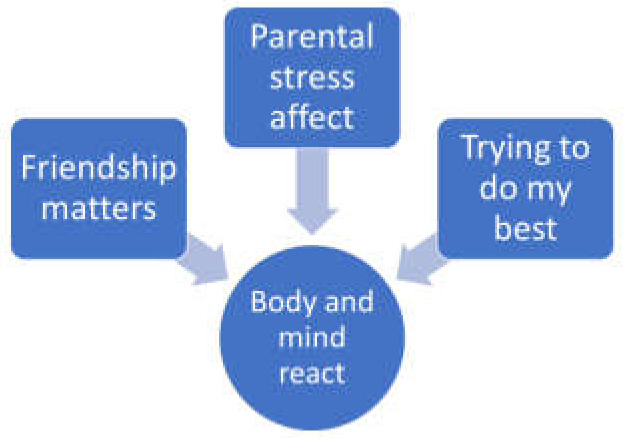
One theme—“Body and mind react”—and three subthemes: (**1**) Friendships matters, (**2**) Parental stress affect, and (**3**) Trying to do my best were identified.

**Table 1 ijerph-17-04021-t001:** Focus group, sex, school, number of participants, and age.

Focus Group (Number)	Sex, Number of Participants	School (Name)	Age (Years)
1	4 girls	A	10
2	3 girls, 1 boy	A	10–11
3	5 girls, 1 boy	A	11–12
4	2 boys	A	11–12
5	1 girl, 2 boys	A	12
6	4 girls	A	11–12
7	4 boys	B	11–12
8	5 girls	B	11–12
9	6 boys	B	11–12
10	4 boys	B	11–12

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
