# Peer review of "A Focus Group Interview Study of the Experience of Stress amongst School-Aged Children in Sweden"

_ijerph, 2020, doi:10.3390/ijerph17114021_

Round 1

Reviewer 1 Report

Aim is clear, including how study was done and what was found. Title is clear, informative and current but could include the area mentioned “, …Municipality, Sweden”. The referencing style seems inconsistent, however, and it is suggested that the authors look at the spacing between numbers especially, such as 2020, 5(15), p. 18 - 22.

The introduction/background information is insufficient. The research question is not outlined in the introduction and could maybe be included again. Furthermore, there is not enough information on past studies or reference to studies involving stress in children aged 10 – 12. A cursory search revealed more than just one study – referred to under “2. Purpose”. More recent studies on stress in early adolescent children should be included to reflect on what is known about the topic already.

Clear processes were outlined in methodology and the research question seemed justified. Reflect on permission obtained from principal. (It is suggested that line 70 – 73 should be moved or to be included in line 104 under 3.5).

Participant ID is confusing with referencing style. In order to make this clearer, perhaps use “Participant 1 or P1”, as opposed to numbers which might be confused with citations.

This is a Qualitative study and the themes are appropriately identified, as well as what is statistically significant.

I do feel that the Results and Discussion sufficiently addressed the research questions and therefore valuable.  I do however, feel that point 7 (Implications for practice) and 8 (Conclusion) lack integration into the study and should include more information regarding findings in older studies.

Author Response

Response to Reviewers

A focus group interview study of experience of stress amongst school-aged children in Sweden

Thank you for your thoughtful comments. We have responded to each comment in italics to facilitate recognition. The changes in the manuscript are marked with yellow

Reviewer 1

Aim is clear, including how study was done and what was found. Title is clear, informative and current but could include the area mentioned “, …Municipality, Sweden”.

Response: The title has now been modified: A focus group interview study of experience of stress amongst school-aged children in Sweden.

The referencing style seems inconsistent, however, and it is suggested that the authors look at the spacing between numbers especially, such as 2020, 5(15), p. 18 - 22.

Response: We have now checked the references.

The introduction/background information is insufficient. The research question is not outlined in the introduction and could maybe be included again.

Response: We have now included the research question in the introduction.

Furthermore, there is not enough information on past studies or reference to studies involving stress in children aged 10 – 12. A cursory search revealed more than just one study – referred to under “2. Purpose”. More recent studies on stress in early adolescent children should be included to reflect on what is known about the topic already.

Response: We have now added more recent studies on stress in early adolescent children.

Clear processes were outlined in methodology and the research question seemed justified. Reflect on permission obtained from principal. (It is suggested that line 70 – 73 should be moved or to be included in line 104 under 3.5).

Response: The text about permission from the principal and the information sent to the parents is now moved to the section 3.5: Ethical consideration.

Participant ID is confusing with referencing style. In order to make this clearer, perhaps use “Participant 1 or P1”, as opposed to numbers which might be confused with citations.

Response: The participant ID is now changed to Focus Group No. X

This is a Qualitative study and the themes are appropriately identified, as well as what is statistically significant.

I do feel that the Results and Discussion sufficiently addressed the research questions and therefore valuable.  I do however, feel that point 7 (Implications for practice) and 8 (Conclusion) lack integration into the study and should include more information regarding findings in older studies.

Response: We have now modified the point 7 (Implications for practice) and 8 (Conclusion).

Reviewer 2 Report

Really appreciate the “giving voice to children” beyond a likert scale. This study gives us depth in understanding. 

I believe the lit review could benefit from some theory application in which youth development is framed.  I’m thinking about this especially when I read the results section of youth picking up on adults’ stress (system theory).  I see this throughout the results section.  Then the discussion could highlight a theory you see as relevant.  

I like the idea of stress being motivational for youth.  Important to differentiate from negative stress. 

I think there could be more discussion related to reducing school stress by teaching more about self regulation and specifically, social emotional learning.  This has been shown to also reduce incidences of bullying. 

I have made comments throughout the pdf for your consideration. 

Author Response

Response to Reviewers

A focus group interview study of experience of stress amongst school-aged children in Sweden

Thank you for your thoughtful comments. We have responded to each comment in italics to facilitate recognition. The changes in the manuscript are marked with yellow

Reviewer 2

Really appreciate the “giving voice to children” beyond a likert scale. This study gives us depth in understanding.

I believe the lit review could benefit from some theory application in which youth development is framed.  I’m thinking about this especially when I read the results section of youth picking up on adults’ stress (system theory).  I see this throughout the results section.  Then the discussion could highlight a theory you see as relevant. 

Response: We have now added some theory application.

I like the idea of stress being motivational for youth.  Important to differentiate from negative stress.

I think there could be more discussion related to reducing school stress by teaching more about self regulation and specifically, social emotional learning.  This has been shown to also reduce incidences of bullying.

Response: Thanks for this comment. We have now added more discussion related to reducing school stress.

I have made comments throughout the pdf for your consideration.

Response: Thank you very much. We have changed the manuscript according to your suggestions.

Reviewer 3 Report

Learning from pre-adolescents/early adolescents about their lived experiences is important to our understanding of their bio-psycho-social functioning and wellbeing in the larger context of the family life  and peer relationships.  The authors are to be commended for designing and executing this study to achieve a deeper understanding their perspectives.

To enhance the strength of the manuscript, the following comments and questions are offered:

Stress in children as a research topic has been well studied at least using quantitative methods, and perhaps not as much using qualitative methods in Sweden as indicated by the authors who found one previous study where children were interviewed about stress.  To strengthen the background  and purpose sections of the paper, it would be important for the authors to provide a brief summary of the recent quantitative studies on stress and children this age group, and what is lacking or is informing from those studies that the study conducted by the authors addresses or fills a gap in understanding about children’s understanding and experiences with stress.

The methods are fairly well presented. Adding a few more details would provide greater clarity. Specifically, the authors mention use of an inductive qualitative design. A brief description and definition of the inductive method would as well as the rationale for its use would be helpful; especially since readers may  be curious as to why some other qualitative approach  was not used to study the phenomenon, as it has been studied little and not much is known due to few published studies.

The authors mentioned that the 1st two authors read the transcripts independently and then they came together for the next phase of the analysis. Did the 1st two authors develop their codes independently at any point prior to coming together and examine inter rater reliability? The authors mention that codes were compared based on differences and similarities-was this a comparison of codes from each of the authors?

Is it safe to assume that the authors were not the moderator or observer in the interviews conducted?

Did the focus groups consist of both boys and girls or were separated by gender?

What was the rationale for including the pilot interviews with the post-pilot interviews? Was it because no changes were made to the questions?  Did the two pilot interviews include one with the children from the rural areas and the other from the urban areas?

The authors state that they strategically selected two schools in two municipalities, urban and rural , with high and low parental employment. Was any data analysis attempted specific to each group (urban and high employment; rural and average employment)?  Since the results are presented for the whole group, can it be concluded that analysis revealed no differences so that all the data could be combined?  

Were the interviews analyzed by gender?

Some of the results of the study were similar to previous findings from the study by Brobeck et al., so in what ways does this study’s findings expand our understanding of stress experienced by children ages 10-12.     

If it is accurate, the interviews were conducted by persons knowledgeable about interviewing and interviewing children of this age group. 

The sample included  a diverse group of children in regards to geography and economic conditions.

The authors can also speak to the strength and limitations of not being directly involved in the interviews in regard to principles and philosophy of using qualitative methods.

The results revealed an important role played by friends and friendships in mitigating/buffering the effects of stress. The authors should discuss what implications this has for practice; in particular for designing prevention and education programs not only for children but also for educators and parents.

Line 25:  consider parents employment instead of employment among parents

Line 31: consider  the ages of 10-12 instead of age 10-12

Line 56:  provide citation in regard to “there is a call for studies…”  If this is the authors conclusion based on review of the literature, then it should be stated as such as a need instead of “call for studies…”

Author Response

Response to Reviewers

A focus group interview study of experience of stress amongst school-aged children in Sweden

Thank you for your thoughtful comments. We have responded to each comment in italics to facilitate recognition. The changes in the manuscript are marked with yellow

Reviewer 3

Learning from pre-adolescents/early adolescents about their lived experiences is important to our understanding of their bio-psycho-social functioning and wellbeing in the larger context of the family life  and peer relationships.  The authors are to be commended for designing and executing this study to achieve a deeper understanding their perspectives. To enhance the strength of the manuscript, the following comments and questions are offered:

 Stress in children as a research topic has been well studied at least using quantitative methods, and perhaps not as much using qualitative methods in Sweden as indicated by the authors who found one previous study where children were interviewed about stress.  To strengthen the background  and purpose sections of the paper, it would be important for the authors to provide a brief summary of the recent quantitative studies on stress and children this age group, and what is lacking or is informing from those studies that the study conducted by the authors addresses or fills a gap in understanding about children’s understanding and experiences with stress.

 Response: Thank you very much for your comments. We have now added a brief summary of recent studies on stress and children.

The methods are fairly well presented. Adding a few more details would provide greater clarity. Specifically, the authors mention use of an inductive qualitative design. A brief description and definition of the inductive method would as well as the rationale for its use would be helpful; especially since readers may  be curious as to why some other qualitative approach  was not used to study the phenomenon, as it has been studied little and not much is known due to few published studies.

Response: We have now added information about the rational for using an inductive method.

The authors mentioned that the 1st two authors read the transcripts independently and then they came together for the next phase of the analysis. Did the 1st two authors develop their codes independently at any point prior to coming together and examine inter rater reliability? The authors mention that codes were compared based on differences and similarities-was this a comparison of codes from each of the authors?

Response: The analysis is now clarified.

Is it safe to assume that the authors were not the moderator or observer in the interviews conducted?

Response: All authors took turns in being moderator and observer. This is now clarified.

Did the focus groups consist of both boys and girls or were separated by gender?

Response: The focus groups consisted of both boys and girls in three cases. Table 1 does now clarify this.

 What was the rationale for including the pilot interviews with the post-pilot interviews? Was it because no changes were made to the questions?  Did the two pilot interviews include one with the children from the rural areas and the other from the urban areas?

Response: The pilot interviews were conducted at the urban school. The rationale for including the pilot interviews in the study is now clarified.

The authors state that they strategically selected two schools in two municipalities, urban and rural , with high and low parental employment. Was any data analysis attempted specific to each group (urban and high employment; rural and average employment)?  Since the results are presented for the whole group, can it be concluded that analysis revealed no differences so that all the data could be combined? 

Response: We did not attempt to specific each group. The qualitative content analysis focus on the variations in the data. All data were combined in the analysis.

Were the interviews analyzed by gender?

Response: The interviews were not analyzed by gender.

Some of the results of the study were similar to previous findings from the study by Brobeck et al., so in what ways does this study’s findings expand our understanding of stress experienced by children ages 10-12.    

Response: We have now clarified in what ways our study expand the understanding of stress experiences by children aged 10-12.

If it is accurate, the interviews were conducted by persons knowledgeable about interviewing and interviewing children of this age group.

The sample included  a diverse group of children in regards to geography and economic conditions.

The authors can also speak to the strength and limitations of not being directly involved in the interviews in regard to principles and philosophy of using qualitative methods.

Response: The authors were involved in the interviews.

The results revealed an important role played by friends and friendships in mitigating/buffering the effects of stress. The authors should discuss what implications this has for practice; in particular for designing prevention and education programs not only for children but also for educators and parents.

Response: Thank you for this comment. We have now added a discussion about what implication this has for practice.

Line 25:  consider parents employment instead of employment among parents

Response: We have changed the wording according to your suggestion.

Line 31: consider  the ages of 10-12 instead of age 10-12

Response: We have changed the wording according to your suggestion.

Line 56:  provide citation in regard to “there is a call for studies…”  If this is the authors conclusion based on review of the literature, then it should be stated as such as a need instead of “call for studies…”

Response: We have changed the wording according to your suggestion.

Reviewer 4 Report

REVIEWER (s)'s COMMENTS to the Editor and Authors:              

Journal: International Journal of Environmental Research and Public Health - ijerph-813034 -

Title: “A focus group interview study of the experience of stress amongst school-aged children “ 

Research original paper qualitative study: 

Keywords: stress; school-aged children; experience; focus group; interviews; qualitative study

Summary to the editor and authors:

Thank you very much for giving me the opportunity to review this manuscript. This is a very interesting paper addressing an important topic in the field of getting an insight into experience of amongst school-aged children. The title sounds interesting and I would like to encourage the authors to revise this manuscript for minor changes.

From my point of view, it is a very well-structured article which focuses on a relevant topic in stress research from children’s perspective by using adequate methods to answer the research question. This article brings up a relevant topic to the readers as researchers, health care professionals, and to institutions and public authorities for educating children.

The authors were concentrated on to write an interesting article by using established methods and to present a relevant selection of the literature which the authors included critically and seriously. The introduction is well structured. The authors were able to represent a complex study in a condensed article including specific quotations in the discussion section. They considered the guidelines of the journal and identified and addressed a research gap in describing relevant topics from children’s perspective, relevant to health care and further research.

Nevertheless, I would like to motivate the authors to include some aspects and arguments to strengthen this relevant article. For this review I follow the COREQ guideline and will relate to some aspects focusing on the methods parts (please see attachment fulfilled).

The manuscript is very well readable und understandable.

This manuscript is convincing in its clarity, well written, easy to read and understand. The methods used are appropriate to the research question and the results address the research question. The discussion is substantively written, incorporating relevant and current evidence. The limitations are understandable and credibly justified.

Please, consider, I am not a native speaker and my English language competences are limited. Based on this fact I am not able to provide substantial feedback on a language specific evaluation. From my point of view as a not native speaker is this paper very well written and clearly to read and to understand.

I would like to motivate the authors to provide some arguments to following aspects:

Introduction part

Please connect your results to existing stress models or concepts.

Methods part:

Methods are well done, grounded on suitable methods, provided literature supports the applied proceedings.

Please provide a short statement or reason about methodology why you have choosen focus groups and not individual face to face interviews.

Line:  64 Focus group interviews were conducted to achieve a deeper understanding 64 of the phenomenon (stress) through interaction in the group of children (12). Focus groups provides a broader insight into the spectrum of arguments of perspective, individual interviews provide a deeper insight into the interviewee’s world. Did you really mean deeper or broader?

Line: 79/80 Please provide information about the background role of researchers who has conducted the focus groups. «Each focus group interview was conducted by one moderator and one observer, with experience talking with children 80 in this age group.» It has an influence on the data process and analysis.

Please describe how the training was done within the two researchers how did the analysis.

  • Have you worked with memos –
  • Do you use a software for analysis the data (MAXQDA, NVIVO, Atlas, others)

Please provide a table with participants characteristics (age, gender, years of education, ….)

Please Provide a figure to present your results it is more obviously to detect the main categories.

Author Response

Response to Reviewers

A focus group interview study of experience of stress amongst school-aged children in Sweden

Thank you for your thoughtful comments. We have responded to each comment in italics to facilitate recognition. The changes in the manuscript are marked with yellow

Reviewer 4

Summary to the editor and authors:

Thank you very much for giving me the opportunity to review this manuscript. This is a very interesting paper addressing an important topic in the field of getting an insight into experience of amongst school-aged children. The title sounds interesting and I would like to encourage the authors to revise this manuscript for minor changes.

From my point of view, it is a very well-structured article which focuses on a relevant topic in stress research from children’s perspective by using adequate methods to answer the research question. This article brings up a relevant topic to the readers as researchers, health care professionals, and to institutions and public authorities for educating children.

The authors were concentrated on to write an interesting article by using established methods and to present a relevant selection of the literature which the authors included critically and seriously. The introduction is well structured. The authors were able to represent a complex study in a condensed article including specific quotations in the discussion section. They considered the guidelines of the journal and identified and addressed a research gap in describing relevant topics from children’s perspective, relevant to health care and further research.

Nevertheless, I would like to motivate the authors to include some aspects and arguments to strengthen this relevant article. For this review I follow the COREQ guideline and will relate to some aspects focusing on the methods parts (please see attachment fulfilled).

The manuscript is very well readable und understandable. This manuscript is convincing in its clarity, well written, easy to read and understand. The methods used are appropriate to the research question and the results address the research question. The discussion is substantively written, incorporating relevant and current evidence. The limitations are understandable and credibly justified.

Please, consider, I am not a native speaker and my English language competences are limited. Based on this fact I am not able to provide substantial feedback on a language specific evaluation. From my point of view as a not native speaker is this paper very well written and clearly to read and to understand. I would like to motivate the authors to provide some arguments to following aspects:

Introduction part. Please connect your results to existing stress models or concepts.

Response: Thank you very much for your comments. We have now connected our result with existing stress models/ concepts.

Methods part: Methods are well done, grounded on suitable methods, provided literature supports the applied proceedings.

Please provide a short statement or reason about methodology why you have choosen focus groups and not individual face to face interviews.

Response: We have now added a short statement why focus groups were chosen.

Line:  64 Focus group interviews were conducted to achieve a deeper understanding 64 of the phenomenon (stress) through interaction in the group of children (12). Focus groups provides a broader insight into the spectrum of arguments of perspective, individual interviews provide a deeper insight into the interviewee’s world. Did you really mean deeper or broader?

Response: We have changed the wording according to your suggestion.

Line: 79/80 Please provide information about the background role of researchers who has conducted the focus groups. «Each focus group interview was conducted by one moderator and one observer, with experience talking with children 80 in this age group.» It has an influence on the data process and analysis.

Response: We have now added information about the background role of researchers.

Please describe how the training was done within the two researchers how did the analysis.

Response: The analysis process is now described more in detail.

Have you worked with memos –

Response: Yes, we worked with memos. This is now clarified.

Do you use a software for analysis the data (MAXQDA, NVIVO, Atlas, others)

Response: No, we did not use a software for analysis of the data.

Please provide a table with participants characteristics (age, gender, years of education, ….)

Response: We have now added a table with participant characteristics.

Please Provide a figure to present your results it is more obviously to detect the main categories.

Response: We have now added a figure to present the result.